# *In silico* mutagenesis of human ACE2 with S protein and translational efficiency explain SARS-CoV-2 infectivity in different species

**Javier Delgado Blanco**[1], **Xavier Hernandez-Alias**[1], **Damiano Cianferoni**[1], **Luis Serrano**[1,2,3]*

**1** Centre for Genomic Regulation (CRG), The Barcelona Institute of Science and Technology, Dr. Aiguader 88, Barcelona, Spain, **2** Universitat Pompeu Fabra (UPF), Barcelona, Spain, **3** ICREA, Pg. Lluís Companys 23, Barcelona, Spain

* luis.serrano@crg.eu

**Data Availability Statement:** The software used in this study is available at http://foldxsuite.crg.es and https://github.com/hexavier/SARSCoV2_species. The authors declare that the data supporting the

## Abstract

The coronavirus disease COVID-19 constitutes the most severe pandemic of the last decades having caused more than 1 million deaths worldwide. The SARS-CoV-2 virus recognizes the angiotensin converting enzyme 2 (ACE2) on the surface of human cells through its spike protein. It has been reported that the coronavirus can mildly infect cats, and ferrets, and perhaps dogs while not pigs, mice, chicken and ducks. Differences in viral infectivity among different species or individuals could be due to amino acid differences at key positions of the host proteins that interact with the virus, the immune response, expression levels of host proteins and translation efficiency of the viral proteins among other factors. Here, first we have addressed the importance that sequence variants of different animal species, human individuals and virus isolates have on the interaction between the RBD domain of the SARS-CoV-2 spike S protein and human angiotensin converting enzyme 2 (ACE2). Second, we have looked at viral translation efficiency by using the tRNA adaptation index. We find that integration of both interaction energy with ACE2 and translational efficiency explains animal infectivity. Humans are the top species in which SARS-CoV-2 is both efficiently translated as well as optimally interacting with ACE2. We have found some viral mutations that increase affinity for hACE and some hACE2 variants affecting ACE2 stability and virus binding. These variants suggest that different sensitivities to coronavirus infection in humans could arise in some cases from allelic variability affecting ACE2 stability and virus binding.

## Author summary

In these early stages of the COVID-19 pandemic it is urgent to understand all features determining the new virus expansion. Two significant factors conditioning infection are ACE2-mediated SARS-CoV-2 cellular entry and viral proteome translation efficiency. Genomic variability across species, including humans, results in ACE2 variants that destabilize its fold, modify ACE2/SARS-CoV-2 recognition, or both. We also point out the

findings of this study are available within the paper and its Supporting Information files.

**Funding:** We acknowledge the support of the Centre for Genomic Regulation (CRG) Technology & Business Development Office (TBDO) for support with licensing information, the CRG Tecnologías de Información y Comunicación (TIC) for assistance with web hosting, and the Scientific Information Technologies (SIT) for distributed computing, the Spanish Ministry of Science and Innovation (MICINN), 'Centro de Excelencia Severo Ochoa', the CERCA Programme/Generalitat de Catalunya, the Spanish Ministry of Science and Innovation (MICINN) to the EMBL partnership. The project that gave rise to these results was supported by a fellowship from "la Caixa" Foundation (ID 100010434; fellowship code LCF/BQ/DI19/11730061). The work of X.H. has been supported by a PhD fellowship from the Fundación Ramón Areces.

**Competing interests:** The authors have declared that no competing interests exist.

importance of considering waters at the interface of protein-protein interactions when performing *in silico* mutagenesis.

## Introduction

In December 2019, the first patients with symptoms of atypical pneumonia were detected in Wuhan (China) [1]. Since then, the coronavirus disease COVID-19 has already caused over 1 million deaths worldwide (as of October 4th 2020), constituting the most severe pandemic of the last decades [2]. The etiologic agent of the outbreak is the novel betacoronavirus SARS-CoV-2, which potentially emerged in a zoonotic jump from another species [3]. Among possible species, previous reports define bats as the likeliest natural host of its SARS-CoV-2 progenitor [4]. In fact, the bat coronavirus RaTG13 is the phylogenetically closest strain to SARS-CoV-2 [5]. In terms of translational adaptation, the codon usage of the new coronavirus is most similar to some birds and mammals [6]. Whether the putative zoonotic jump occurred directly from bats or through other intermediate species remains still elusive. In an attempt to identify such intermediate species in close contact with humans, a recent study shows that ferrets and cats are highly susceptible to SARS-CoV-2 [7]. There are recent reports suggesting that dogs can be infected as well [8] while livestock, including pigs, chickens, and ducks, as well as mice and rats are not susceptible to infection.

A range of several factors can determine the tropism of a virus, which includes (1) the mechanism of viral entry into the host, (2) the hijack of cellular machinery to support viral replication, (3) the translational efficiency of the viral proteins and (4) the ability to elude the immune response [9]. Recent reports have shown 119 host proteins associated with different coronavirus that play a role in its replication [10] and recent two-hybrid analysis has found 251 host proteins targeted by SARS-CoV-2 [11], and 332 in a pull-down experiment [12]. Despite this rich interactome information and the existence of several viral structures, we only have structural information on the complex between the spike protein (S) and the host receptor angiotensin converting enzyme 2 (ACE2), which determines the cellular entry of the virus in the cell [13]. Up to the time the manuscript was written (May 2020), there is an electron microscopy structure of the complex of the SARS-CoV-2 S protein with the neutral amino acid transporter $B^0AT1$ and the soluble part of human ACE2 (hACE2)[14]. There are are two crystal structures of the RBD domain of the S protein of SARS-CoV-2 and hACE2 (6moj [15]; 6lzg [16], as well as one of a chimeric RBD (SARS-CoV/SARS-CoV-2) with hACE2 (PDB id: 6vw1 [17]). Aside from these complexes, there is structural information of the interaction between the RBD domain of the S protein of other coronavirus and ACE2 from different hosts [16,18–20]. In these studies, it was mentioned that the major species barriers are determined by interactions between four ACE2 residues (residues 31, 35, 38, and 353) and two RBD residues (residues N479 and T487) [19]. Supporting the idea that the interaction of the S protein with ACE2 is critical for virus infection, it has been shown that, by changing four residues on the surface of rat ACE2 to human, rats can be infected by the SARS-CoV [21]. In this work, they identified residues 82–84 and 353 of ACE2 as critical for interaction with the S protein of this virus. Similarly, changing residues K479 and S487 in civet SARS-CoV S protein to N479 and T487 significantly enhanced the binding affinity for hACE2 [19].

A second factor that is important for virus infectivity is the adaptation of the viral codon usage to that of its host [22,23]. The universal genetic code indicates that multiple 3-letter combinations of nucleotides can encode for the same amino acid (aka synonymous codons). However, these different synonymous codons can be recognized distinctly by cellular tRNAs,

leading to differences in translational efficiency [24]. In particular, in terms of translational adaptation of SARS-CoV-2 to human tissues, the viral proteome is especially adapted to the tRNA levels of the upper respiratory tract and the lung parenchyma [23]. This is also in agreement with single-cell transcriptomics describing ACE2 expression in nasal goblet and ciliated cells as well as type-2 alveolar epithelial cells [25,26]. In concordance, patients of SARS-CoV-2 showcase high viral loads in nasal swabs compared to other tissues of the respiratory tract [27].

Here, we have analyzed two of factors that could affect the infectivity of the SARS-CoV-2 in different species as well as the possible sensitivity of humans with different ACE2 variants. First, we looked at the complex between the hACE2 enzyme and the RBD domain of the virus S protein. We modeled with FoldX [28] the ACE2 amino acid variants at the interface with the S protein that are found in different species compared to hACE2. To do so, we first predicted water bridges between residues at the interface which are important for complex stability and specificity [29,30]. Then we determined the binding energy of the modeled variants. We found that the RBD domain of SARS-CoV-2 can recognize the ACE2 from ferrets, civets, cats, and dogs, but not that of pigs, chicken, ducks, mice and rats. Second, we found out the high translational adaptation of SARS-CoV-2 in *Homo sapiens*, compared to other species, which could explain its high infectivity in humans.

Then we looked at the variability reported for hACE2 as well as for different viral isolates (covid19beacon.crg.eu). There is evidence that some people can be infected with no apparent symptoms [31–33], whereas at the same time apparently healthy young people could have serious infections [34]. While this could be due to many factors, it is also quite possible that genetic variants of the ACE2 protein could exhibit different affinities for the virus [35]. In fact, there are actual differences in distribution and allele frequencies of expression quantitative trait loci for ACE2 in different populations [36]. We found two human variants that could affect the interaction with the S protein, increasing or decreasing the susceptibility to infection. We also find that many of the human variants could significantly destabilize the ACE2 protein and therefore reduce active expression at the surface of the target lung cells, which could also affect sensitivity to infection. Finally, we modeled all single point mutations in the RBD domain of the S protein, predicting the effect on binding energy with hACE2 and S protein stability. Our results agree with previous predictions on the importance of residue N501 [37], highlighting the danger of potential mutations occurring in the S protein.

## Results

### Structural description of the ACE2-S protein complex

There are three crystal structures of the hACE2 soluble part with a domain of the S protein of the SARS-CoV-2 (PDB id: 6vw1 2.68 Å resolution, 6lzg 2.5 Å resolution, 6m0j 2.45 Å resolution). 6vw1 is a crystallographic dimer of a chimeric RBD domain of SARS-CoV and SARS-CoV2 with hACE2, containing 2 slightly different binding conformations (6vw1_1, 6vw1_2). The three X-ray structures are very similar, superimposing with a maximum RMSD of 1.88Å over 738 aligned residues (S5 Table). hACE2 contacts the S protein through two separate regions leaving a central cavity that must be filled with water molecules (Fig 1A). Using a simple 4.5Å contact distance cut-off, the residues involved on ACE2/S interface are hACE2-Q24 (sc-sc H-bond with S-N487), hACE2-T27 (hydrophobic packing with S-F456, S-Y473, S-A475, and S-Y480), hACE2-F28 (Van der Waals' contact with S-Y489), hACE2-D30 (sc-sc H-bond with S-K417), hACE2-K31 (hydrophobic packing with S-L455, S-F456, and S-Y489; salt bridge with S-E484; weak sc-sc h-bond S-Q493), hACE2-H34 (Van der Waals contact with S-Y453, and S-Q493), hACE2-E35 (sc-sc H-bond with S-Q493), hACE2-D38 (sc-sc H-bond with S-Y449), hACE2-Y41 (sc-sc H-bond with S-T500), hACE2-Q42 (sc-sc H-bond with S-Y449,

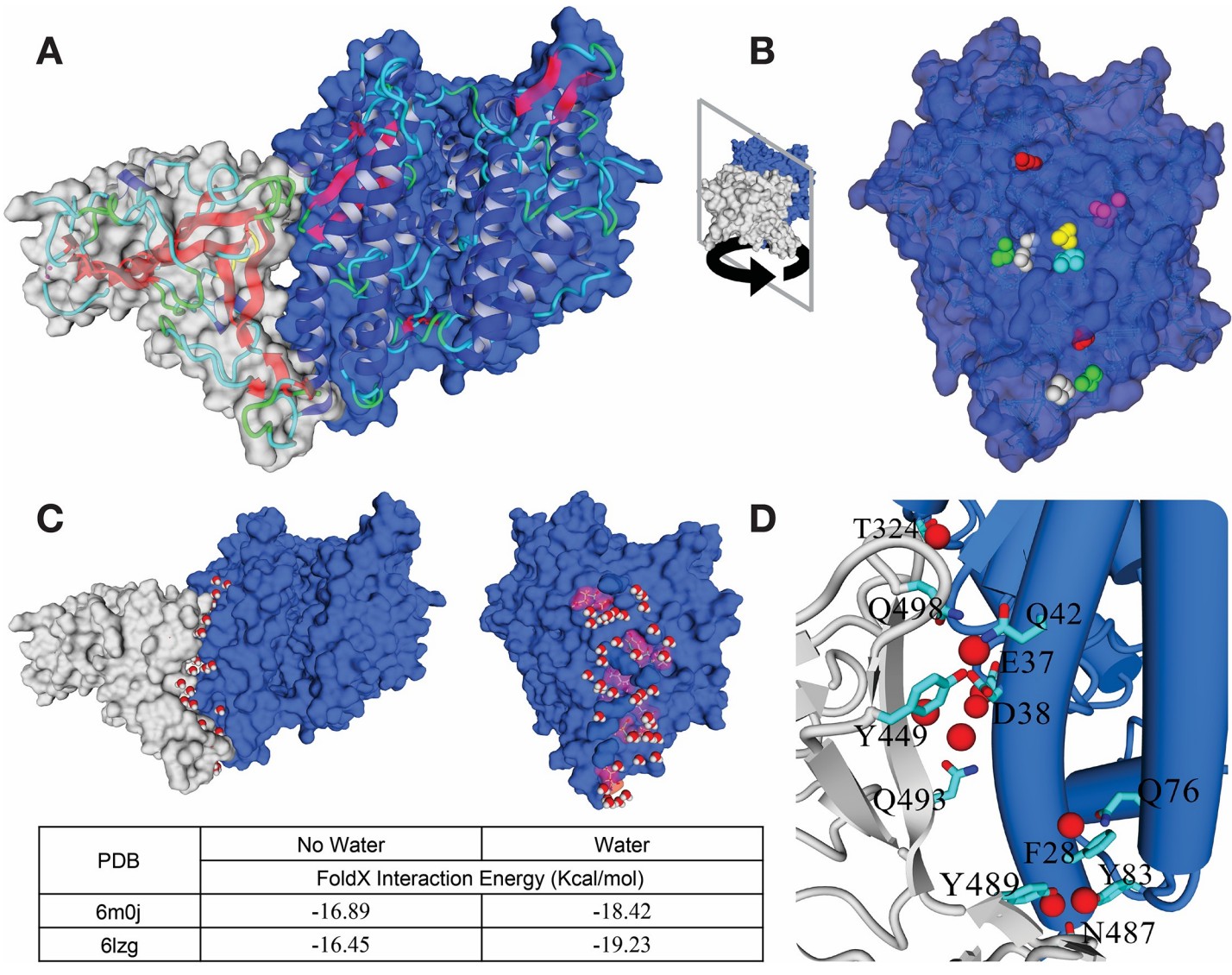

| PDB | No Water | Water |
|---|---|---|
| | FoldX Interaction Energy (Kcal/mol) | |
| 6m0j | -16.89 | -18.42 |
| 6lzg | -16.45 | -19.23 |

**Fig 1. Structural description of the ACE2-S protein complex.** (A) Complex between the soluble ACE2 protein and a domain of the S protein from the virus (S1 domain grey/ACE2 blue). (B) ACE2 surfaces for all models described (6m0j, 6lzg, 6vw1, and civet-human ACE2 chimaera: 3doi), 9 predicted conserved water clusters are shown in different colors. (C) Water prediction over the ACE2/SARS-CoV-2 S protein contact surface using 6m0j structure and contribution to binding energies as determined by FoldX. (D) Atomic detail of 6m0j with the water molecules corresponding to the nine conserved clusters showing the protein residues interacting with them. S1 domain (backbone in gray) and ACE2 (backbone in blue). In S6 Table we show the coordinates of the predicted water molecules for complex 6moj.

and S-Q498; sc-mc h-bond with S-Y446), hACE2-L45 (hydrophobic packing with S-V445, and S-Q498), hACE2-L79 (hydrophobic packing with S-F486, and S-Y489), hACE2-M82 (hydrophobic packing with S-F486), hACE2-Y83 (sc-sc H-bond with S-N487; weak sc-sc H-bond with S-Y489; PI-PI interaction with S-F486), hACE2-Q325 (hydrophobic packing with S-V503), hACE2-N330 (Van der Waals contact with S-T500), hACE2-K353 (hydrophobic packing with S-Y505; sc-mc H-bond with S-G496), hACE2-D355 (weak sc-sc H-bond with S-T500), hACE2-D357 (weak sc-sc H-bond with S-T500) and hACE2-D393 (weak sc-sc H-bond with S-V503). Water molecules bound at the interface of protein-protein interactions play an important role in affinity and specificity [29,30]. In fact, looking at the structure of the complex we find many instances of side chains from the two molecules capable of donating

and/or accepting H-bonds that are close in space but not in contact (Fig 1B). These residues could interact via a water molecule (water bridge). One of them in structure 6lzg is bridging K353 of hACE2 with the side chain of N501 and the main chain carbonyl of G496 in the virus (S1A Fig). This is interesting since K353 in hACE2 and N501 in the virus are key residues in the interaction between the two proteins [15] but actually they don't directly contact the other protein in this 6lzg or 6m0j. Since the number of water molecules at the interface of the three proteins is very low probably because of their medium crystallographic resolution, we used the protein design algorithm FoldX to predict water bridges at the interface. FoldX has been shown to predict crystal water bridges with extreme accuracy [38]. FoldX recapitulates 100% of the crystallographic water bridges in the three structures and predicts new water bridges at the surface of the two proteins, filling the interface and connecting residues from the two proteins (Fig 1C). An example involves hACE2 R357 and hACE2 D355. These residues have been previously described to be in Van der Waals contact with T500 of the S protein using a cutoff of 4.5Å [16]. This cutoff is very tolerant to consider them in direct contact and in fact they interact via a water bridge (D355 side chain with T500 carbonyl group and R357 side chain with T500 side chain oxygen (S1B Fig)). Notably, we found 9 water clusters at the interface of ACE2 and the RBD domain of the S-protein in all three structures (Fig 1B). These water bridges expand the connectivity between interface residues (Fig 1B and 1D) and can contribute individually up to 0.6 kcal/mol, and overall up to 2.78 kcal/mol (see 6moj in Fig 1).

Prior to any mutation modeling, we first looked at the PDB structure quality parameters (www.rcsb.org; S5 Table) to decide which structure is better for modelling. We found that 6m0j is the best by all criteria. This structure has only 0.1% of the residues in disallowed areas of the Ramachandran plot, followed by 6lzg that has 0.4% of disallowed residues. Structure 6vw1 aside from being a chimaera has 0% of the residues in disallowed areas but the lowest scores for the quality parameters. Thus we decided to use 6m0j as our structure to model the different variants in hACE2 and the RBD domain of the S protein (we also did the same mutations in 6lzg, see Discussion and S1–S5 Tables; with one exception discussed below there is an excellent correlation for the changes in energy upon mutation in both structures).

## Modeling the binding affinity of ACE2 from different animals

Once having a repaired PDB (6moj) with water bridges, we could proceed to model the ACE2 of different animal species selected for this study. To do so, first we had to ensure that the variants found in different species, compared to the hACE2, do not introduce conformational changes which could preclude using the complex between hACE2 and the RBD domain of the S protein as a model. First, we checked that there are no insertions and deletions between the ACE2 sequences corresponding to the region binding the RBD domain (Fig 2A). Second, we superimposed the hACE2 interface region with the chimeric human-civet one (3doi) showing that they are identical within the range of crystallographic errors (S2 Fig). Third, mutation of residues in the hACE2 interface region, required to adapt hACE2 to the species analyzed here, shows no incompatible substitution with the hACE2 structure (considering an error of 0.8 kcal/mol in absolute terms [39], in FoldX predicted values; see S2 Fig). The only exception is position 79 in chicken, duck and mouse (S2 Fig). But this residue is solvent exposed and therefore should not compromise the structure. Water prediction in the civet structure displays also the 9 clusters described above, which is another indication of the conservation of sidechain packing at the ACE2/S binding surface. Thus we could reasonably assume that the differences in ACE2 interface residues among different species does not significantly change the complex and therefore existing hACE2-RBD structures could be used as a template to model the ACE2 from other species.

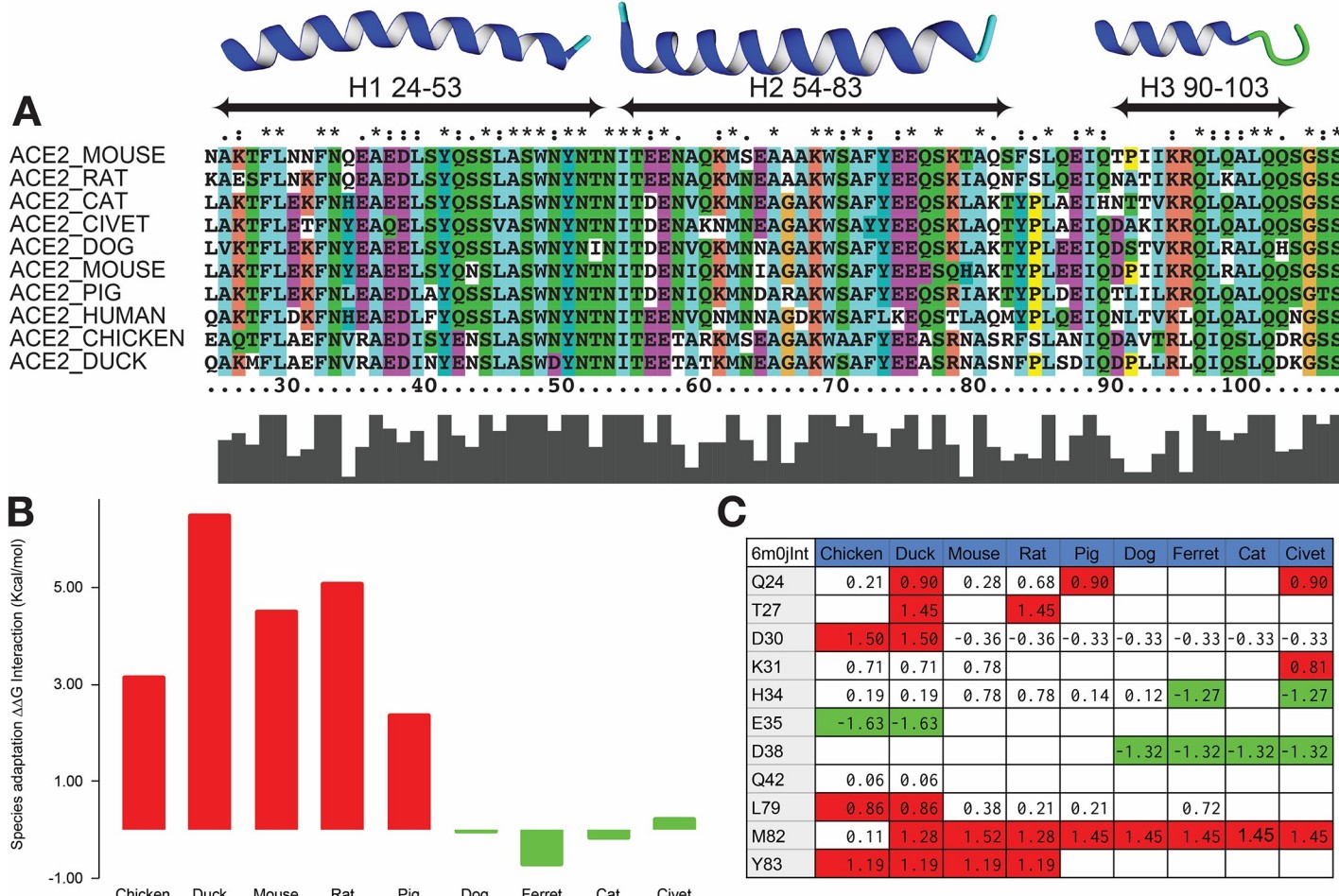

**Fig 2. Binding affinities of animal species.** (A) ACE2 full sequence alignment of the selected species for the binding region to the S protein (B) Global ΔΔG interaction for different species by adding single residue contributions with respect to hACE2. Green bars for species susceptible to be infected, red bars for species not infected. (C) Per-residue interaction ΔΔG values in kcal/mol with respect to the hACE2 residues. We don't show the results for K353 in this table since it is not on the helical interface and it is only mutated to His in mice.

Then we mutated each of the positions that differ between hACE2 and the other ACE2 sequences located at the interface of the complex (Fig 2A) and determined the changes in binding energy (see Materials and Methods and S1 Table). In Fig 2B we show the difference in binding energy between the ACE2 of the species analyzed here and the S protein. In Fig 2C we show the effects of individual mutations. Mutation of K31 (a critical residue for binding [20]) to Glu (in *Gallus gallus*) or to Asn (in *Mus musculus*) breaks two charged hydrogen bonds with the side chain of Q493 and the F490 backbone oxygen of the S protein destabilizing the complex by 1.9 kcal/mol. M82 in human hACE2 is also important and its mutation to Thr or Asn is quite destabilizing. The same happens with Y83 that makes a side chain H-bond with Asn487 from the S protein. As reported, K353 is another hot spot forming a hydrogen bond with G496 backbone oxygen of the S protein [20].

Overall we see a very good agreement between the changes in binding energy with respect to hACE2 and the infectivity of the virus [40]. Dog, cat, civet and ferret ACE2 have comparable interactions energies as hACE2, and all have been reported to be infected by the virus. The remaining species considered in this study present worse interaction energies and are, in fact, not infected [40].

## SARS-CoV-2 translational efficiency and ACE2 expression across species

While the infectivity of different species can be explained by the ACE2-S protein binding affinity, this alone cannot completely explain the severity of the disease in each species. Upon the productive interaction of the viral spike glycoprotein and the cell receptor ACE2, the viral genome enters the cell and starts its replication. The coronavirus therefore needs to hijack the translational machinery of the host to efficiently replicate and produce new virions. In this context, the codon usage of viral proteins should potentially resemble that of the host cell in order to adapt to the tRNA pools that drive an optimal translation [23].

The proteome of SARS-CoV-2 is mainly composed of the replicase polyprotein (ORF1ab) and of structural proteins: the spike glycoprotein, the membrane and envelope proteins, and the nucleoprotein [41]. Based on the genomic codon usage of each of the possible host species, we compute the codon adaptation index (CAI) and the tRNA adaptation index (tAI) to estimate the translational efficiency of SARS-CoV-2 proteins in each host (Fig 3A and 3B and S2 Table). Humans are among the top three species whose CAIs are mostly over 0.70, together with ducks and and chicken. In terms of the tAI, humans show the highest translational adaptation among all others, followed by chicken, and, to some extent, mice and rats. On the other hand, cats, ferrets, pigs, and dogs are less translationally adapted than humans both by CAI and tAI.

Together with the translational adaptation of the viral proteome, another factor determining the viral infectivity to host cells is the ACE2 receptor expression [43]. Comparing the expression levels of ACE2 across species, we see that chicken and dogs have the lowest expression (Fig 3B and S2 Table). However, an expression level as low as in humans might be sufficient to infect the lung. Although previous reports indicate that SARS-CoV-2 is also able to infect the upper respiratory tract in humans [23,27,40], gene expression data of that tissue is not available for most other species.

Overall, ACE2 relative expression at RNA level does not seem to explain infectivity by the virus. This could be because RNA levels do not always correlate well with protein expression [44]. CAI and tAI as estimates of translational efficiency could better explain, together with binding affinity, species infectivity. We observed that humans are the species in which SARS-CoV-2 is both most efficiently translated as well as optimally interacting with ACE2.

## ACE2 human variation and interaction with S protein

In an attempt to explain differences in infection sensitivity, we looked for human missense point mutations for the ACE2 gene in the Ensembl database [45] plus dbVar [46]. We found a total of 260 reported single point mutations (S3 Table, which also includes allelic frequencies). We mutated each of the positions in 6moj using FoldX and determined the changes in stability of the ACE2 protein, as well as in interaction energy with the S protein (S3 Table) for mutations having significant effects, with more than 0.8 kcal/mol in absolute terms [39]. We find only one variant G326E that significantly improves binding energy without destabilizing the hACE2 protein (Fig 4 and S3 Table). There are two mutations that consistently decrease binding without affecting hACE2 stability and could confer protection (E37K, T27A). We also found that 110 out of 260 mutations destabilize significantly the ACE2 protein (>1.5 kcal/mol), which could prevent its correct folding and therefore the binding to the virus (S3 Table).

## S protein point mutation energy landscape and interaction with ACE2

Using both 6m0j and 6lzg structures, we generated each of the 20 possible mutations for all RBD residues (S4 Table). The percentage of mutations that destabilize the spike protein is

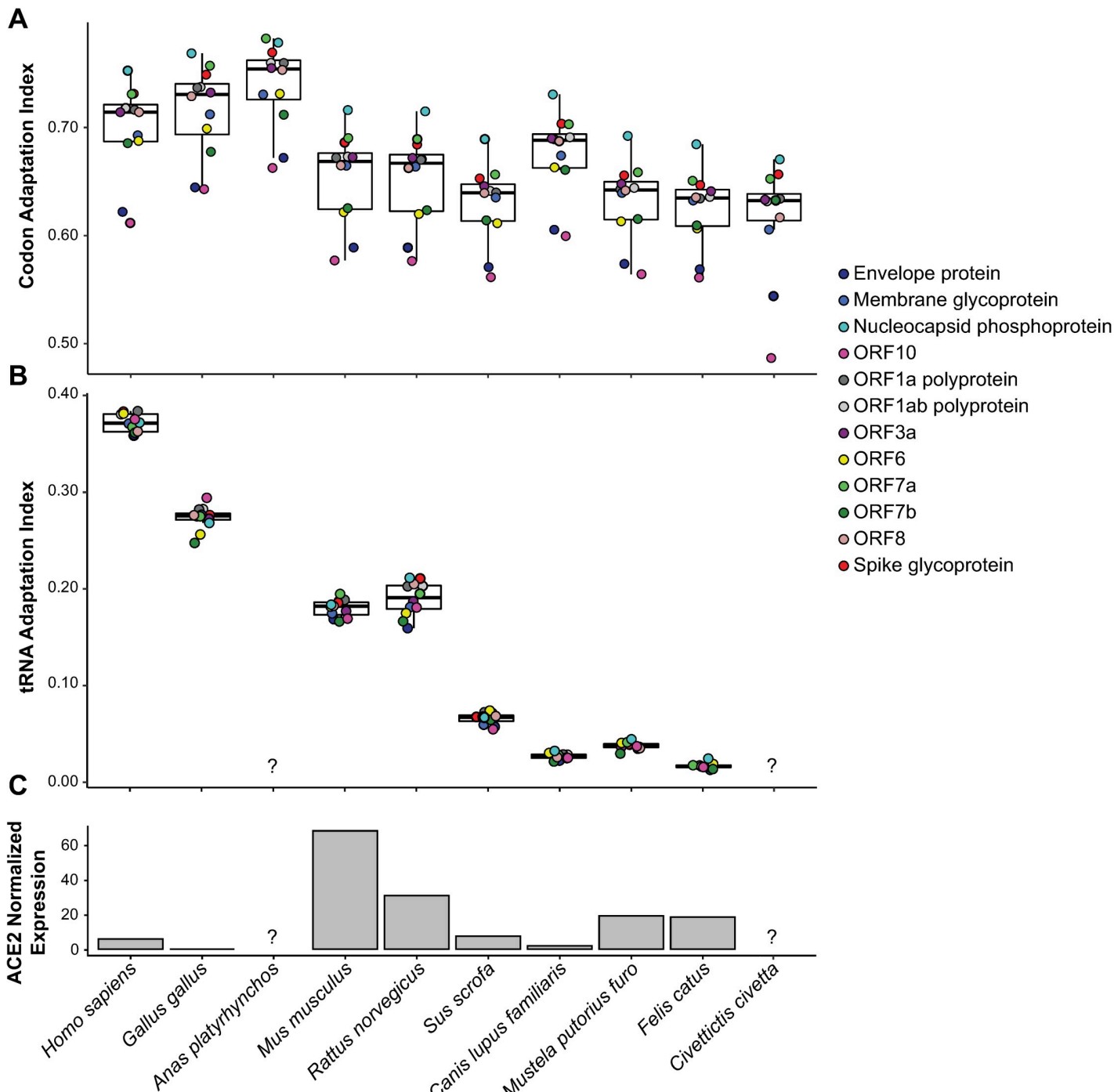

**Fig 3. SARS-CoV-2 translational efficiency and ACE2 expression across species.** (A) Codon Adaptation Index (CAI) of all viral proteins across different species. (B) tRNA Adaptation Index (tAI) of all viral proteins across different species. Boxes expand from the first to the third quartile, with the center values indicating the median. The whiskers define a confidence interval of median ± 1.58*IQR/sqrt(n). (C) Median ACE2 gene expression in lung tissues of each species, normalized by the housekeeping gene ACTB. RNA-seq expression levels were retrieved from Sun et al. (2020) or from the Bgee database [42].

~43%, while ~1% stabilizes it. Considering the interface residues of the complex, we found that ~35% of the mutations would decrease binding affinity of the complex and only ~6% of them would improve it.

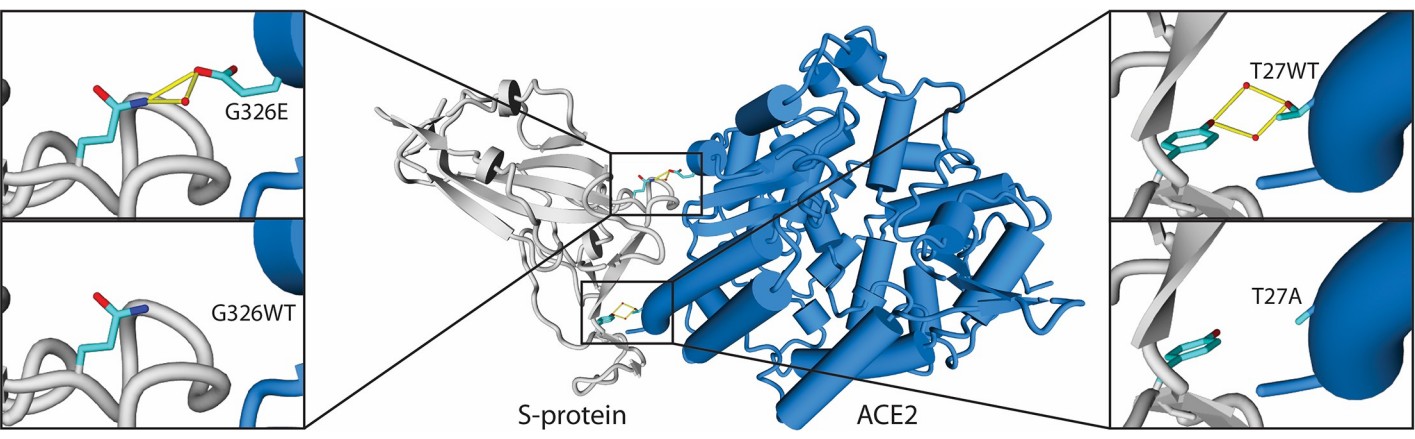

**Fig 4. ACE2 human variants that affect ACE2/SARS-CoV-2 S protein complex interaction energy for 6m0j.** G326E increases hACE2 affinity for S protein whereas T27A decreases it by means of H bonding and water bridge creation or deletion without having a significant change in hACE2 stability.

We observed that several mutations on five interface residues (V445M, V445R, V445W; Q493F, Q493L, Q493M, Q493Y; Q498F, Q498L, Q498M, Q498Y; T500K; N501A, N501C, N501L, N501S, N501T, V503R, 503W, V503Y) were beneficial for the interaction with hACE2 without destabilizing the S domain. Based on the coding sequence of S protein, we observe that, out of the 20 favourable amino acid substitutions, 4 would require at least one nucleotide mutation, 9 at least two mutations, and the remaining 7 three nucleotide mutations. We compared our mutational interaction energy landscape of S protein with a list of 3773 observed missense variants resulting in 2420 unique amino acid mutations found in S protein gene using the CRG Viral Beacon (covid19beacon.crg.eu). Among the observed natural mutations we detected 6 of them predicted as detrimental for the interaction with hACE2: L455F, A475V, K417N, N487K, Y489H, A475S (S4 Table) and we did not observe any of the ones that improve binding.

Residues Q493 and N501 belong to a group of six interface residues (L455, F486, Q493, S494, N501, Y505) fundamental for binding to ACE2 receptors and for determining the host range of SARS-CoV-like viruses [3]. Although the effect of multiple mutations on interaction energy is not necessarily additive when making a single multiple mutant, we observe that mutating the 6 positions mentioned to their relative ones in SARS-CoV S protein lowers the interaction affinity by +5.2 kcal/mol in 6m0j. Interestingly, residue N501 corresponds to a Thr in SARS-CoV S protein and we find that, in agreement with previous predictions [37], mutating N501 to Thr improves binding to hACE2 (S4 Table).

## Discussion

The coronavirus SARS-CoV-2, causant of the deadliest pandemic of the last decades, has most likely appeared upon a zoonotic transfer from another animal host to humans [3]. However, how the putative progenitor strain evolved until acquiring the high infectivity of SARS-CoV-2 is still unknown. In this study, we wanted to explain the susceptibility of different animals by analyzing an ensemble of infection determinants as a whole, as well as the effect of hACE2 variants in binding to the S viral protein. To do so, we first repaired the structures of the complexes between both proteins adding water molecules to the interface using FoldX [28]. The deposited crystallographic structures have a central cavity at the interaction surface filled by water and we predict many water molecules bridging side chains from the two proteins that contribute to the overall interaction (around 3 kcal/mol). Before mutating the residues at the

complex interface which are different between hACE2 and the other animal ACE2 proteins, we examined if these sequence differences could result in conformational changes. Then we proceeded to introduce the different animal variants at the interface in an individual manner. We could do this because the differences are located dispersed over the binding surface far away from each other. We see a nice correspondence between the observed infectivity of animal species and the binding energy predicted by FoldX, with ferrets, cats, dogs, and civets having similar interaction energies to humans. However, it is fundamental to point out the importance of using the best crystal structure. Although we see an excellent correlation between the effect of mutating residues of hACE2 in the 6moj and 6lzg structures, there is one case where the result is very different. This happens at position D38 where mutation to Glu is favourable in 6m0j and unfavourable in 6lzg (S4 Table). The reason for this difference is the network of H-bonds made by the side chains of hACE2 Q42, Q498 and Y449, which is incorrect in 6lzg. The reason is that in the 6lzg structure the proton of the OH side chain group of Y449 (S protein) is donated to the hACE2 D38 carboxylate group, as a result the O of the Tyr449 OH group is at H-bond distance of the oxygen of the CO side chain group of Q498 (hACE2) which is not possible (S1C and S1D Fig). FoldX cannot repair this because there is a double reciprocal H bond between the side chains of Q42 and Q498 in hACE2 and therefore it does not move them. This does not happen in 6m0j where all H-bonds are correct and allow substitution of D38 by Glu without destabilizing the complex. Thus it is important to examine the quality of the structures prior to the *in silico* mutagenesis. In any case, and for information purposes, we include the same data presented in this work for 6m0j and for 6lzg which are in very good agreement except for a few cases as the one mentioned here (S1 Table and S4 Table).

While binding affinity to ACE2 could justify the infectivity of SARS-CoV-2 across species, it alone failed to explain the severity of infections compared to humans [7]. For this reason, we additionally took the translational adaptation and ACE2 expression into consideration. While it seems that a low expression of ACE2 in lungs is sufficient to produce infection, the CAI and tAI across species could explain some of the previous concerns. In particular, dogs, pigs, cats, ferrets and civets show all a poorer translational efficiency than humans, explaining why SARS-CoV-2 produces the most severe infections in the latter. This is in concordance with recent findings showing that viruses resemble the codon usage of symptomatic hosts more than that of non-symptomatic counterparts [47]. On the other hand, we find that chicken and ducks have a good CAI and tAI which could allow efficient virus replication, but as indicated above they have a very bad binding energy to ACE2. This is important because it is easier to select for a few mutations at the interface of the interaction than to change the CAI, and therefore the virus could jump to these species.

Overall, by concurrently interrogating the binding affinity to ACE2 and the translational adaptation of SARS-CoV-2, we could explain the susceptibility of animal species to viral infection. We have also identified three human variants that could increase or decrease viral susceptibility by affecting the interaction of the two proteins, and a large number of human variants that destabilize ACE2. We have also examined the effects in stability and interaction energies for all possible variants of the S protein interface residues which could be useful when finding new viral missense mutations. In this respect, we have found some possible mutations that could increase the binding energy for hACE2. Understanding the grounds of infectivity will be essential to develop targeted therapies and identify possible intermediate hosts and vectors of this virus.

## Materials and methods

### Side chain mutagenesis and energy calculations

ACE2 stability and hACE2/SARS-CoV-2 S protein interaction free energies upon mutation ($\Delta\Delta G$ kcal/mol) were computed for interface residue positions using two crystallographic

complexes (PDB ids: 6m0j:A/B; 6lzg:A/B). Side chain modeling of these positions to all 20 standard amino acids was carried out using the FoldX BuildModel command after repairing crystallographic defects using the RepairPDB command both for the complex and for the naked ACE2. The interaction energy was calculated using the FoldX AnalyseComplex command. The global energies of the animal species were calculated by adding the mean contribution for the three models' corresponding mutations. Water prediction was done using the FoldX CrystalWaters command [38]. Bound metals were considered by using the FoldX CrystalMetal command. The same procedure was applied to model the human variants. FoldX user manuals for all commands can be found at http://foldxsuite.crg.es. A document containing all necessary descriptions to run the calculations is included (S1 Text).

We determined the changes in free energy upon mutation of the two crystal complexes and determined the average energy change and standard deviation (S1 Table). Only mutations where we have a significant free energy change in two of the three structures (>0.8 or <-08 kcal/mol for binding and >1.5 and <-1.5 for stability kcal/mol) were considered.

FoldX does not recognize the glycosylated Asn residues and therefore we did not compute the changes in stability or binding when the mutation involves one of these residues or if the mutated position interacts with them.

## Codon Adaptation Index (CAI)

The CAI is an estimate of translational efficiency based on the similarity of codon usage with regard to a reference set of genes [48]. The rationale is that a coding sequence is optimized when it uses the same codons as highly expressed genes do. In our case, we compare the viral genes against the whole genome of each species. The codon usage tables of species from RefSeq were downloaded from the Codon/Codon Pair Usage Tables (CoCoPUTs) project release as of April 3, 2020 [49].

The first step is to compute a reference table of normalized codon usage for each species, which is defined as the genomic abundance of a certain codon compared to the most abundant synonymous codon. These weights are determined by dividing the frequency of each codon $f_c$ by the maximum frequency among all codons within each amino acid family.

$$w_c = \frac{f_c}{max_{i \in c_{aa}}(f_i)}$$

The CAI of a certain protein is the product of weights $w$ of each codon $i_k$ at the triplet position $k$ throughout the full gene length $l_g$, and normalized by the length.

$$CAI = (\prod_{k=1}^{l_g} w_{i_k})^{1/l_g}$$

The coding sequences of SARS-CoV-2 coronavirus were retrieved from the reference genome and annotations at RefSeq (NC_045512.2).

## 1.1 tRNA Adaptation Index (tAI)

The tAI is an estimate of translational efficiency based on the correspondence between the codon usage of genes and the tRNA copy numbers (i.e. a coding sequence is optimal when it uses the codons for which a high number of tRNA genes are present in the genome). As described by [50,51], the tAI weights every codon based on the wobble-base codon-anticodon interaction rules. Let $c$ be a codon, then the decoding weight is a weighted sum of the square-root-normalized tRNA abundances $tRNA_{cj}$ for all tRNA isoacceptors $j$ that bind with affinity

$(1-s_{cj})$ given the wobble-base pairing rules $n_c$. The pairing affinity of each codon-anticodon is therefore defined by a set of s-values that is specific to each species. The species-specific decoding weights of each codon, based on the tRNA copy numbers of their genomes and the corresponding s-values [52], were downloaded from the STADIUM database as of June 23, 2020 [53].

$$w_c = \sum_{j=1}^{n_c}(1 - s_{cj})tRNA_{cj}$$

The tAI of a certain protein is the product of weights $w$ of each codon $i_k$ at the triplet position $k$ throughout the full gene length $l_g$, and normalized by the length.

$$tAI = (\prod_{k=1}^{l_g}w_{i_k})^{1/l_g}$$

The coding sequences of SARS-CoV-2 coronavirus were retrieved from the reference genome and annotations at RefSeq (NC_045512.2).

## ACE2 gene expression

The ACE2 normalized expression in lung tissue of cats, dogs, ferrets, and pigs was directly retrieved from a previous study [43]. For chicken, humans, mice, and rats, the ACE2 expression of lungs in FPKM (Fragments per Kilobase Million) was downloaded from the Bgee database [42]. We then used the house-keeping gene ACTB to normalize gene expression across species, reproducing the same analysis as in previous studies [43].

$$ACE2\ normalized\ expression = \frac{FPKM_{ACE2}}{FPKM_{ACTB}} \cdot 10000$$

## Allelic frequencies of human variants

We used the Ensembl database [45] (gnomAD, TopMed, ExAc, ESP, 1000Genomes) and dbVar to extract missense human variations and the allelic frequencies of those reported human mutations generating point amino acidic variants of ACE2. For each allelic variant its frequency was retrieved from the largest study between the available ones that determined each single SNP and using the global population subset.

## Supporting information

**S1 Fig. Important interacting residues detailed.** (A) Water bridge between hACE2-K353, S-N501, S-Q498, S-G496 (B) Hbond network and water bridges for hACE2-R357. (C) Hbond network around D38 in the 6moj structure. (D) Hbond network around D38 in the 6lzg structure, it can be seen how the O of the OH group of Tyr449 is at H-bond distance to the CO group of Gln 498 which is not possible.
(TIF)

**S2 Fig. Structural and energetic analysis of the ACE2 binding interface in different species.** (A) Superimposition of the three alpha-helices (24–53,54–83,90–103) contacting the S protein from all X-ray structures (6lzg, 6m0j, 6vw1-2 crystallographic dimers, 3doi-2 crystallographic dimers). (B) Local sequence alignment of the ACE2 residues that are in the region of the ACE2 that contacts against the S protein. We show those that are different between the species. (C) Changes in folding stability of the hACE2 protein upon single point mutations from human to

the animal species using either the 6lzg or 6moj structures.
(TIF)

**S1 Table. ACE2/SARS-CoV-2 S protein FoldX ΔΔG Interaction PSSM tables for contacting residues.** Energies calculated using the 6m0j and the 6lzg structures. All units are in kcal/mol.
(XLS)

**S2 Table. Codon Adaptation Index and tRNA Adaptation Index of SARS-CoV-2 proteins and ACE2 expression in lung tissues across species.** Related to Fig 3.
(XLSX)

**S3 Table. ACE2 Stability (kcal/mol), Interaction energies (kcal/mol) and Allele frequencies for Human Variants.** Energies calculated for models: 6lzg, 6m0j.
(XLS)

**S4 Table. S protein point mutation energy landscape and interaction with ACE2.** Contains stability (of the S protein alone) and interaction energy (of the complex) predictions for all possible point mutations of S protein, computed using both 6m0j and 6lzg crystallographic complexes. Each sub-table includes a summary of the mutation energy distribution with respect to the sensitivity of the FoldX force field.
(XLSX)

**S5 Table. Quality Control for crystallographic ACE/S structures.** It includes PDB quality control parameters, FoldX stability energy and all against all global RMSDs for 6m0j, 6lzg, 6vw1 (two crystallographic dimers) and 3doi (two crystallographic dimers).
(XLSX)

**S6 Table. Water prediction for 6m0j.** It includes crystallographic and predicted waters coordinates for 6m0j.
(XLSX)

**S1 Text. FoldX commands used for calculations.** It includes the commands used for running repair PDB, ACE2/S interaction energy, ACE2 stability, and water prediction.
(PDF)

## Author Contributions

**Conceptualization:** Luis Serrano.

**Formal analysis:** Javier Delgado Blanco, Xavier Hernandez-Alias, Damiano Cianferoni.

**Funding acquisition:** Javier Delgado Blanco, Luis Serrano.

**Investigation:** Javier Delgado Blanco, Xavier Hernandez-Alias, Damiano Cianferoni.

**Methodology:** Javier Delgado Blanco, Xavier Hernandez-Alias, Damiano Cianferoni, Luis Serrano.

**Project administration:** Javier Delgado Blanco.

**Software:** Javier Delgado Blanco, Xavier Hernandez-Alias, Damiano Cianferoni.

**Supervision:** Luis Serrano.

**Validation:** Javier Delgado Blanco, Xavier Hernandez-Alias, Damiano Cianferoni.

**Visualization:** Javier Delgado Blanco, Xavier Hernandez-Alias, Damiano Cianferoni.

**Writing – original draft:** Javier Delgado Blanco, Xavier Hernandez-Alias, Damiano Cianferoni, Luis Serrano.

**Writing – review & editing:** Javier Delgado Blanco, Xavier Hernandez-Alias, Damiano Cianferoni, Luis Serrano.

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
