## [Decision Letter · Decision Letter 0]

16 Jun 2020

Dear Dr. Serrano,

Thank you very much for submitting your manuscript "ACE2 genetic variability and codon usage explains coronavirus infectivity in species suggesting different resistance degrees in humans" for consideration at PLOS Computational Biology.

As with all papers reviewed by the journal, your manuscript was reviewed by members of the editorial board and by several independent reviewers. In light of the reviews (below this email), we would like to invite the resubmission of a significantly-revised version that takes into account the reviewers' comments.

We cannot make any decision about publication until we have seen the revised manuscript and your response to the reviewers' comments. Your revised manuscript is also likely to be sent to reviewers for further evaluation.

Sincerely,

Rachel Kolodny

Guest Editor

PLOS Computational Biology

Nir Ben-Tal

Deputy Editor

PLOS Computational Biology

Reviewer's Responses to Questions

**Comments to the Authors:**

Reviewer #1: Title: ACE2 genetic variability and codon usage explains coronavirus infectivity in species suggesting different resistance degrees in humans

Summary: The authors investigate the infectivity of SARS-CoV-2 through computational analysis of several different topics. They begin with modeling the human ACE2 – SARS-CoV-2 S protein complex using the software FoldX in order to predict the presence of water bridges that could contribute to stronger binding between the two molecules. They proceed to model the same complex using amino acid residues from homologues ACE2 in other potential host animals at the ACE2 – S complex interface. They compare the codon adaptation index of the viral genes in human and animals as well as the lung tissue-specific expression of ACE2 in each system. They end with an analysis of 229 reported SNPs in the ACE2 receptor and their potential impact on the binding of the ACE2 – S protein complex. While the manuscript provides some interesting modeling, the different analyses that are conducted are not inherently linked and the conclusions that are drawn are not well supported. In order to be considered for publication, the study should unify the message of these varied investigations, either increasing focus on ACE2 – S protein interactions with a more robust computational analysis or considering other host interactions with the viral S protein.

We offer the following comments:

1. As the manuscript does not go into extensive detail regarding the ACE2 – S protein interaction beyond what has heretofore been published, the authors could expand their analysis beyond viral protein interactions with the ACE2 receptor. Zhou et al. (2020) find 119 host proteins associated with coronaviruses, including SARS-CoV-2. If the authors intend to include an analysis of other species and their ACE2 receptors, they should consider expanding the analysis to include the known interactome.

2. The authors analyze the binding energies between the SARS-CoV-2 S protein and ACE2 receptor in human and other animals. They rely on the published crystal structure of human ACE2 – S protein complex and introduce amino acid mutations where the animal residues are different at the interface of the complex. This might be sufficient but docking (with a tool such as ZDOCK) is more realistic. Docking may find that this orientation isn’t achievable due to clashing in intermediate orientations in the docking process. Furthermore, the assumption that ACE2 in other animals will have the same structure is not well based. Other parameters should be taken in consideration such as conservation, mRNA structure, etc.

3. The authors relate the infectivity of SARS-CoV-2 to the CAI of its genes in human and other potential hosts. They also attempt to draw a connection between ACE2 expression in host lung tissue and infectivity. It is noted that SARS-CoV-2 infects cats, yet the authors report some of the lowest CAIs for viral genes in Felis catus. They also report one of the lowest ACE2 expression levels in human lung. These contradictory analyses do not add to the message of the manuscript. CAI is a metric of codon usage bias and is not directly linked to translation efficiency. Additional approaches should be taken to estimate translation efficiency, otherwise this part should be eliminated altogether. The low expression of ACE2 in human lung and high expression in mouse lung (which is not infected by SARS-CoV-2) is not explained in the context of the study. In general, this is the weakest part of the study, it does not have novelty, while the conclusions are only loosely supported by the data.

4. The authors analyze 229 mutations in the human ACE2 receptor. While they utilize the Single Nucleotide Polymorphism Database (dbSNP), they could improve the power of this analysis by including data from NCBI’s database of human genomic structural variation (dbVAR).

In its current state, the manuscript does not gain from analysis of CAI and ACE2 expression in other species. These investigations could be interesting if made more thorough. Currently the authors offer evidence that is contradictory to the message of the study. It is recommended that the authors either expand these analyses with more robust metrics and experimental data if these are available or instead focus on the ACE2 – S protein interaction in human, including other proteins from the interactome and a more comprehensive analysis of variation in human ACE2.

Reviewer #2: uploaded as an attachment

**Have all data underlying the figures and results presented in the manuscript been provided?**

Reviewer #1: Yes

Reviewer #2: Yes

PLOS authors have the option to publish the peer review history of their article (what does this mean?). If published, this will include your full peer review and any attached files.

Reviewer #1: No

Reviewer #2: No
---

## [Decision Letter · Decision Letter 1]

7 Sep 2020

Dear Dr. Serrano,

Thank you very much for submitting your manuscript "In silico mutagenesis of human ACE2 with S protein and translational efficiency explain SARS-CoV-2 infectivity in different species" for consideration at PLOS Computational Biology.

As with all papers reviewed by the journal, your manuscript was reviewed by members of the editorial board and by several independent reviewers. In light of one of the reviews (below this email), we would like to invite the resubmission of a significantly-revised version that takes into account the reviewers' comments.  Please address these comments.  

We cannot make any decision about publication until we have seen the revised manuscript and your response to the reviewers' comments. Your revised manuscript is most likely to be sent to reviewers for further evaluation.

[Dear Luis, because reviewers tend to 'worn out' (as may have happened with Reviewer 1 here) it may well be that your manuscript will be reviewed by a third reviewer. Please take it into consideration -Nir]

Sincerely,

Rachel Kolodny

Guest Editor

PLOS Computational Biology

Nir Ben-Tal

Deputy Editor

PLOS Computational Biology

Reviewer's Responses to Questions

**Comments to the Authors:**

Reviewer #1: The authors have addressed our comments and suggestions.

Reviewer #2: uploaded as an attachment

**Have all data underlying the figures and results presented in the manuscript been provided?**

Reviewer #1: Yes

Reviewer #2: Yes

PLOS authors have the option to publish the peer review history of their article (what does this mean?). If published, this will include your full peer review and any attached files.

Reviewer #1: No

Reviewer #2: No
---

## [Editor Report · Decision Letter 2]

19 Oct 2020

Dear Dr. Serrano,

We thank you for your detailed response.

We are pleased to inform you that your manuscript 'In silico mutagenesis of human ACE2 with S protein and translational efficiency explain SARS-CoV-2 infectivity in different species' has been provisionally accepted for publication in PLOS Computational Biology.

Best regards,

Rachel Kolodny

Guest Editor

PLOS Computational Biology

Nir Ben-Tal

Deputy Editor

PLOS Computational Biology

---

## [Editor Report · Acceptance letter]

9 Nov 2020

PCOMPBIOL-D-20-00670R2 

*In silico* mutagenesis of human ACE2 with S protein and translational efficiency explain SARS-CoV-2 infectivity in different species

Dear Dr Serrano,

I am pleased to inform you that your manuscript has been formally accepted for publication in PLOS Computational Biology. Your manuscript is now with our production department and you will be notified of the publication date in due course.

With kind regards,

Nicola Davies
